# Exploring the Genotoxic Stress Response in Primed Orphan Legume Seeds Challenged with Heat Stress

**DOI:** 10.3390/genes16020235

**Published:** 2025-02-19

**Authors:** Andrea Pagano, Conrado Dueñas, Nicolò Bedotto, Amine Elleuch, Bassem Khemakhem, Hanen El Abed, Eleni Tani, Maria Goufa, Dimosthenis Chachalis, Alma Balestrazzi

**Affiliations:** 1Department of Biology and Biotechnology ‘L. Spallanzani’, University of Pavia, via Ferrata 9, 27100 Pavia, Italy; andrea.pagano01@unipv.it (A.P.); conradojr.duenas01@universitadipavia.it (C.D.J.); nicolo.bedotto01@universitadipavia.it (N.B.); 2Faculty of Sciences, University of Sfax, Airport Road km 0.5, Sfax P.O. Box 1169-3029, Tunisia; amine.elleuch@fss.usf.tn (A.E.); bassem.khemakhem@fss.usf.tn (B.K.); hanenelabed3@gmail.com (H.E.A.); 3Laboratory of Plant Breeding & Biometry, Department of Crop Science, Agricultural University of Athens, 75 Iera Odos, 11855 Athens, Greece; etani@aua.gr (E.T.); marog@aua.gr (M.G.); 4Benaki Phytopathological Institute, Stefanou Delta Street 8, Kifissia, 14561 Athens, Greece; d.chachalis@bpi.gr

**Keywords:** *Lathyrus sativus* L., *Trigonella foenum-graecum* L., *Pisum sativum* var. *arvense*, seed germination, hydropriming, heat stress, comet assay

## Abstract

Background/Objectives: The increased frequency of extreme weather events related to climate change, including the occurrence of extreme temperatures, severely affects crop yields, impairing global food security. Heat stress resulting from temperatures above 30 °C is associated with poor germination performance and stand establishment. The combination of climate-resilient crop genotypes and tailored seed priming treatments might represent a reliable strategy to overcome such drawbacks. This work explores the potential of hydropriming as a tool to mitigate the heat-stress-mediated impact on germination performance in orphan legumes. Methods: For each tested species (*Lathyrus sativus* L., *Pisum sativum* var. *arvense* and *Trigonella foenum-graecum* L.), two accessions were investigated. Germination tests were performed at 25 °C, 30 °C, 35 °C and 40 °C to assess the heat stress tolerance threshold. Hydropriming was then applied and germination tests were performed at 40 °C to test the impact of the treatment on the seeds’ ability to cope with heat stress. An alkaline comet assay and Quantitative Real Time-Polymerase Chain Reaction were performed on embryos excised from primed and control seeds. Results: Phenotyping at the germination and seedling development stage highlighted the accession-specific beneficial impact of hydropriming under heat stress conditions. In *L. sativus* seeds, the alkaline comet assay revealed the dynamics of heat stress-induced DNA damage accumulation, as well as the repair patterns promoted by hydropriming. The expression patterns of genes involved in DNA repair and antioxidant response were consistently responsive to the hydropriming and heat wave conditions in *L. sativus* accessions.

## 1. Introduction

Heat stress impacts fundamental physiological processes such as photosynthesis and reproductive development, compromising plant performance; thus, mitigation strategies covering short- and medium-term temporal frames are urgently needed [1,2]. The complex molecular networks underlying the response of plants to heat stress still need to be fully elucidated [3,4].

Heat stress is responsible for altered seed quality features, e.g., carbohydrates, proteins, oils, phytochemicals, vitamins, and dietary fibers, resulting from impaired physiological processes, such as source–sink interactions and nutrient transport to the developing seed [5,6]. It has been suggested that the screening of genotypes based on seed germination performance under heat stress may allow information useful to the development of strategic approaches for the identification of superior germplasm to gathered [5]. On the other hand, it is possible to enhance the germination potential under heat stress, using tailored seed-priming techniques [7,8]. These pre-sowing treatments require controlled seed imbibition with water or different priming agents to trigger the transient resumption of the pre-germinative metabolism and boost the seed repair response, namely antioxidant mechanisms and the DNA damage response (DDR) [9]. The treatment must be stopped before radicle protrusion occurs, otherwise seeds will lose their desiccation tolerance, and they will not be able to survive the subsequent dry-back step. Primed seeds show improved germination performance in terms of their germination percentage, speed and synchronicity, as well as increased stress tolerance [9]. Seed priming has been used to alleviate heat stress by inducing the production and activity of heat shock proteins and molecular chaperones, preventing protein denaturation during heat stress conditions [7,8]. The application of brassinosterois as seed-priming agents could promote heat stress tolerance in *Brassica juncea* L. [10]; similar benefits were reported for cytokinins used as a priming agent for wheat seeds [11]. The combined action of sodium nitroprusside and indole acetic acid as priming agents was able to improve heat stress tolerance in tomato (*Solanum lycopersicum* L.) [12].

Orphan legumes are minor crops grown by local populations in the most climate-vulnerable regions, under harsh environmental conditions. These neglected crops, poorly investigated at the genetic level, show unique tolerance profiles against drought, salinity, cold, and heat, combined with a high nutritional value. The hidden potential of orphan legumes has become a precious resource in the context of climate change [13,14,15,16]. Seed priming represents a sustainable strategy for seed quality enhancement in orphan legumes; however, a deeper understanding of the molecular dynamics underlying the seed repair response are required to develop reliable and effective protocols, together with powerful multidisciplinary networks [17].

The DNA damage response plays a critical role in the acquirement of thermotolerance in plants, and it has been reported that the genes responsible for thermotolerance can trigger DDR-mediated processes [18,19,20,21]. An additional level of complexity is the role of the chromatin status in relation to DNA repair capability, in the context of heat stress [22,23,24,25].

In the present work, accessions from the neglected legume species *Lathyrus sativus* L., *Trigonella phoenum-graecum* L., and *Pisum sativum* var *arvense* were investigated under a range of increasing temperatures to assess their tolerance threshold for seed germination. Hydropriming was tested as a tool to improve the seed and seedling response to heat stress. The impact of treatments was assessed, with a focus on the heat stress-dependent DNA damage accumulation profiles and DNA repair, using a alkaline comet assay performed on embryos excised from primed and control seeds. The expression of genes involved in DDR and the antioxidant response has been evaluated as an indicator of the transcriptional response to hydropriming and/or heat stress. The present work is developed using the following hypotheses: (i) different species and accessions of commercially neglected legumes can display different tolerance profiles against heat stress events occurring during pre-germinative stages, (ii) hydropriming can be evaluated as a resource-effective option to improve germination profiles in stressed and/or unstressed legume seeds, and (iii) investigating the seed antioxidant and repair response can present novel directions for understanding the molecular bases of seed vigor in Fabaceae.

## 2. Materials and Methods

### 2.1. Plant Materials, Treatments, and Germination Tests

Seeds of the *Lathyrus sativus* L. accessions Maleme-107 and Sofades were supplied by ELGO-DIMITRA Ellenic Agricultural Organization (Athens, Greece) and Agroland S.A. (Sofades, Greece). Maleme-107 is a commercial variety characterized by autumn sowing, an estimated 1000 seed weight of 185–195 g, a seed yield of 150–220 kg/ha, 16–17 kg/ha sowing, adaptation to dry and hot conditions, and resistance to cold and diseases. Sofades is a local variety characterized by autumn sowing, with an estimated 1000 seed weight of 12–130 g, a seed yield of approximately 150 kg/ha, and 15 kg/ha sowing. Seeds of the *Pisum sativum* var. *arvense* accessions Forrimax and Guifredo were supplied by Semillas Batlle S.A. (Barcelona, Spain). Both Forrimax and Guifredo are nutritional short-term crops that produce forage with high yields, with an estimated 1000 seed weight of 150–180 g and 160–170 g, respectively. Seeds of the *Trigonella foenum-graecum* L. accessions Rayhane and Tborsek were provided by INGC (Institut National des Grandes Cultures, Bou Salem, Tunisia). The Rayhane and Tborsek accessions are cultivated preferentially on drained loamy soil, are adapted to a cold, frost-free climate, and have 12 kg/ha sowing. For germination tests, seeds were transferred into Petri dishes and kept in a growth chamber at 25 °C under light conditions with a photon flux density of 150 μmol m^−2^ s^−1^, a photoperiod of 16/8 h, and 70–80% relative humidity. Seven days after the beginning of the germination test, biometrical analysis was performed by measuring the length of the aerial part and the main root. For each condition, three independent replicates (trays) each containing 20 seeds were considered for statistical analyses. Seeds displaying radicle protrusions (>1 mm) were considered germinated. The germinability (G) and mean germination time (MGT) were calculated according to Ranal and de Santana [26]. Seven days after the beginning of the germination test, the seedling phenotype was classified into three groups based on the following morphological criteria: (i) seedlings displaying the normal development of both the aerial part and main root were classified as normal, (ii) seedlings not displaying the growth of either the aerial part or main root were classified as aberrant, and (iii) seeds not displaying any growth of either the aerial part or main root were classified as non-germinating. Two different experimental systems were designed (Figure 1). For the preliminary screening of heat stress tolerance in terms of germination performance for the target accessions, seeds were imbibed for 4 h at 25 °C; subsequently, a heat wave was applied to test increasing temperatures (30 °C, 35 °C, 40 °C) for 4 h (Figure 1a). This experimental design allowed the most appropriate conditions to be selected for the subsequent analyses in order to assess the impact of hydropriming on heat stress tolerance (Figure 1b). For hydropriming, seeds were imbibed for 8 h and then subjected to dry-back for 24 h. Both unprimed and hydroprimed seeds were subsequently imbibed for 4 h at 25 °C, followed by a heat wave of 40 °C for 4 h, resulting into four treatment groups: (i) unprimed and non-stressed seeds, (ii) unprimed seeds subjected to heat wave, (iii) hydroprimed and non-stressed seeds, and (iv) hydroprimed seeds subjected to heat wave (Figure 1b). For each condition, three independent replicates (trays), each containing 20 seeds, were considered for statistical analyses.

### 2.2. Comet Assay

The alkaline version of the comet assay was used to quantify the single- and double-strand breaks (SSBs, DSBs) formed from alkali-labile sites and DNA–DNA or DNA–protein crosslinks [27]. Seeds from each accession and experimental condition were harvested at the indicated time point (Figure 1b), and embryo axes were isolated from the cotyledons and seed coat using a blade, as reported by Pagano et al. [28]. Nuclei were extracted as previously described [29]. The resulting suspension was mixed in equal volume with a solution containing 1% low-melting-point agarose (Sigma Aldrich, Milan, Italy) in phosphate-buffered saline (PBS: 140 mM of NaCl, 2.7 mM of KCl, 10 mM of Na_2_HPO_4_, 1.8 mM of KH_2_PO_4_) at 38 °C. Three drops of the resulting suspension were pipetted onto agarose-precoated slides and solidified on ice. The slides were incubated for 30 min at 4 °C in alkaline buffer (1 mM of Na_2_ EDTA, 300 mM of NaOH, pH 13.0) and then electrophoresed in the same buffer for 25 min at 0.72 V cm^−1^ in a cold chamber, under dark conditions. After electrophoresis, the slides were washed three times for 5 min using 0.4 M Tris HCl at pH 7.5, rinsed in 70% ethanol (*v*/*v*) three times for 5 min at 4 °C, and dried overnight at room temperature. The slides were stained with 20 μL of 4′,6-diamidino-2-phenylindole (DAPI, 1 mg mL^−1^, Sigma Aldrich). For each slide, 100 nucleoids were scored and visualized using an Olympus BX51 fluorescence microscope (Olympus Italia S.rl., Segrate, Milan, Italy) with an excitation filter of 340–380 nm and a barrier filter of 400 nm. Images were captured using an Olympus MagnaFire camera equipped with Olympus Cell-F Software Version 3.4. The nucleoids were classified as previously described by Collins [27], where each type of nuclei morphology belongs to a class ranging from 0 to 4. The results were expressed in arbitrary units (a.u.), calculated using the following formula: a.u. = [Σ (N_c_ × c) × 100]/N_tot_, where N_c_ is the number of nuclei in each class (c), and N_tot_ is the total number of nuclei counted [27].

### 2.3. RNA Extraction, cDNA Synthesis and Quantitative Real-Time Polymerase Chain Reaction

RNA was isolated from embryo axis tissues sampled from dry seeds, seeds subjected to hydropriming and dry-back, unprimed seeds subjected to 8 h of imbibition at 25 °C, unprimed seeds subjected to 4 h of imbibition at 25 °C followed by 4 h of heat wave at 40 °C, hydroprimed seeds subjected to 8 h of imbibition at 25 °C, and hydroprimed seeds subjected to 4 h of imbibition at 25 °C followed by 4 h of heat wave at 40 °C. Approximately 50 embryo axes were sampled for each condition and stored at −80 °C before RNA isolation. RNA isolation was carried out using the TRIZOL^®^ Reagent (Fisher Molecular Biology, Trevose, PA, USA), according to the supplier’s indications. cDNAs were obtained using the RevertAid First-Strand cDNA Synthesis Kit (Thermo Fisher Scientific, Milan, Italy), according to the manufacturer’s suggestions. Quantitative real-time polymerase chain reaction (*q*RT-PCR) was performed with the Maxima SYBR Green qPCR Master Mix (2×) (Thermo Fisher Scientific), according to supplier’s indications, using a CFX Duet Real-Time PCR System (Bio-Rad, Segrate, Milan, Italy). The amplification conditions were as follows: denaturation at 95 °C for 10 min, and 45 cycles at 95 °C for 15 s, 60 °C for 30 s, and 72 °C for 30 s.

The *L. sativus* reference and target genes were identified from the available *L. sativus* non-annotated transcribed RNA sequences deposited in NCBI Nucleotide (https://www.ncbi.nlm.nih.gov/nuccore/; accessed 20 November 2023) using the Nucleotide BLAST function (https://blast.ncbi.nlm.nih.gov/Blast.cgi; accessed 20 November 2023); this was performed on the *L. sativus* transcriptome shotgun assembly by using the annotated coding sequences in the model as input. *M. truncatula* and *L. sativus* sequences with percent identities >80% were considered orthologs and subsequently utilized for oligonucleotide primer design. Oligonucleotide primers were designed using the Real-Time PCR Primer Design program Primer3Plus (https://www.primer3plus.com, assessed on 19 January 2025) from GenScript and further validated through the online software Oligo Analyzer (https://www.eu.idtdna.com/calc/analyzer, assessed on 19 January 2025) (Appendix A). The following genes were tested (the accession numbers of the *L. sativus* transcriptome shotgun assembly are provided): *LsSODa* (*SUPEROXIDE DISMUTASE*; ortholog a) (GBSS01003115), *LsSODb* (*SUPEROXIDE DISMUTASE*; ortholog b) (GBSS01001045), *LsAPXa* (*ASCORBATE PEROXIDASE*; ortholog a) (GBSS01001446), *LsAPXb* (*ASCORBATE PEROXIDASE*; ortholog b) (GBSS01019041), *LsMT1* (*METALLOTHIONEIN*; type 1) (GBSS01000013), *LsMT2* (*METALLOTHIONEIN*; type 2) (GBSS01000484), *LsSPSD* (*SPERMIDINE SYNTHASE*) (GBSS01000983), *LsSPMS* (*SPERMINE SYNTHASE*) (GBSS01003422), *LsSOG1* (*SUPPRESSOR OF THE RESPONSE 1*) (GBSS01010054), *LsOGG1* (*8-OXOGUANINE GLYCOSYLASE/LYASE 1*) (GBSS01016977), *LsFPG* (*FORMAMIDOPYRIMIDINE-DNA GLYCOSYLASE*) (GBSS01018706), *LsLIG1* (*DNA LIGASE 1*) (GBSS01022069), *LsTOP1* (*DNA TOPOISOMERASE 1*) (GBSS01013769), *LsPCNA* (*PROLIFERATING CELL NUCLEAR ANTIGEN*) (GBSS01017540), *LsTFIIS* (*TRANSCRIPTION ELONGATION FACTOR IIS*) (GBSS01005128) and *Ls5.8SrRNA* (AY839389). *LsEF2* (*ELONGATION FACTOR 2*) (GBSS01000402) and *LsGAPDH* (*GLYCERALDEHYDE-3-PHOSPHATE DEHYDROGENASE*) (GBSS01000223) were selected from a panel of candidate reference genes based on the two lowest M-values using geNorm (https://www.genorm.cmgg.be/, assessed on 19 January 2025) and were subsequently used as reference genes for relative quantitation. The relative quantification of transcript accumulation was performed according to Thomsen et al. [30], using a standardized efficiency (E) value of 1.8. All reactions were carried out in triplicates.

### 2.4. Statistical Analysis

For the preliminary screening of the effects of temperature ranges (25 °C, 30 °C, 35 °C and 40 °C), 3 replicates (trays) per condition and 10 seeds per replicate were used. The statistical significance was assessed through a two-tailed heteroscedastic Student’s *t*-test comparing the heat wave conditions to the optimal unstressed conditions (25 °C). *p*-values < 0.05 were considered statistically significant and indicated with asterisks *, *p*-value < 0.05; **, *p*-value < 0.01; ***, *p*-value < 0.001. For the assessment of the combined effects of priming and heat wave conditions, 3 replicates per condition and 20 seeds per replicate were used, and statistically significant differences (*p*-value < 0.05) were assessed through one-way ANOVA and Duncan’s test, comparing the four treatment groups (unprimed non-stressed seeds, unprimed seeds subjected to heat wave, hydroprimed non-stressed seeds, and hydroprimed seeds subjected to heat wave). qRT-PCR reactions were performed in triplicate and statistically significant differences (*p*-value < 0.05) were assessed through one-way ANOVA and Duncan’s test, comparing the following treatments: seeds subjected to hydropriming and dry-back, unprimed seeds subjected to 8 h of imbibition at 25 °C, unprimed seeds subjected to 4 h of imbibition at 25 °C followed by 4 h of heat wave at 40 °C, hydroprimed seeds subjected to 8 h of imbibition at 25 °C, and hydroprimed seeds subjected to 4 h of imbibition at 25 °C followed by 4 h of heat wave at 40 °C. The publicly available software SumAOV (© 2014 by Houssein I. Assaad; https://houssein-assaad.shinyapps.io/SumAOV/, assessed on 19 January 2025), developed by Assaad et al. [31], was used to perform one-way ANOVA and Duncan’s test. Principal Component Analysis (PCA) was performed using MetaboAnalyst 6.0 (https://www.metaboanalyst.ca/docs/Publications.xhtml, accessed on 20 January 2025) [32], normalizing the raw values by the Z-score (mean-centered and divided by the standard deviation of each variable). The datasets used for PCA analysis included the germination parameters (germination percentage and mean germination time), seedling growth parameters (shoot and root length), and relative gene expression data (for each tested gene, see Section 2.3), collected on the *L. sativus* accessions Maleme-107 and Sofades.

## 3. Results

### 3.1. Threshold for the Heat Stress Tolerance at the Seed Germination Level

The six accessions from the three neglected legume species investigated here, *L. sativus*, *P. sativum* arvense and *T. foenum-graecum*, were first screened for their germination performance under heat stress. According to the experimental system shown in Figure 1a, seeds were imbibed for 4 h at 25 °C and subsequently exposed to heat waves of increasing temperatures (30 °C, 35 °C, and 40 °C) for 4 h. In the case of *L. sativus*, the impact of heat waves on the germination parameters and seedling growth of the Maleme-107 and Sofades accessions is shown in Figure 2. A significant (*p*-value < 0.001) decrease in germinability (G) occurred in both varieties upon exposure to 4 h of heat wave at 40 °C. No significant changes were observed in seeds treated at 30 °C and 35 °C, compared to the control (NT). In response to the highest temperature, Sofades exhibited a higher G value (51.67 ± 7.64) than Maleme-107 (25.00 ± 5.00) (Figure 2a). An increase in the mean germination time (MGT) was detected in both varieties exposed to 4 h of heat wave at 40 °C (68.20 ± 1.93, Maleme-107; 44.06 ± 3.51, Sofades); however, Sofades also showed a decreased MGT value (36.11 ± 2.50) when treated with 4 h of heat wave at 35 °C (Figure 2b). The reported data suggest that Sofades is characterized by a higher level of heat stress resilience in terms of germination performance compared to Maleme-107. The impact of heat stress on seedling development was also evaluated. The frequency of seedlings with a normal phenotype developed from seeds exposed to 4 h of heat wave at 40 °C decreased in both varieties; however, Sofades had a higher frequency (21.67 ± 2.89%) of seedlings with a normal phenotype compared to Maleme-107 (10.00 ± 10.00%) (Figure 2c).

The response of the two *P. sativum* arvense accessions, Forrimax and Guifredo, is shown in Figure 3. When considering the estimated G values, no significant differences were observed for Forrimax seeds treated with heat stress, compared with the control. A similar response was noticed for Guifredo; however, these seeds failed to germinate when exposed to 4 h of heat wave at 40 °C (Figure 3a). A significant (*p*-value < 0.001) increase in MGT was detected in Forrimax seeds exposed to 4 h of heat wave at 40 °C, as well as in Guifredo seeds exposed to 4 h of heat wave at 35 °C (Figure 3b). Based on the reported data, Guifredo seeds show a lower tolerance threshold than Forrimax. The frequency of seedlings with a normal phenotype developed from seeds exposed to 4 h of heat wave at 40 °C decreased in Forrimax (25.00 ± 5.00%) compared to NT and the other treated samples. Similarly, Guifredo had a higher frequency (63.33 ± 5.77%) of seedlings with a normal phenotype, in response to the 4 h heat wave at 35 °C (Figure 3c).

Finally, the two *T. foenum graecum* accessions, Rayhane and Tborsek, showed different responses to heat stress (Figure 4). A decrease in germinability was observed in both varieties exposed to 4 h of heat wave at 40 °C. No significant differences were detected in their response to heat waves at 30 °C and 35 °C. At 40 °C, Tborsek showed an estimated G value (46.67 ± 5.77) higher than that observed for Rayhane (26.67 ± 5.77) (Figure 4a). No significant changes in the mean germination time (MGT) were observed in response to heat stress (Figure 4b). Based on germination performance, Rayhane turned out to be more sensitive to heat stress than Tborsek. No seedlings with a normal phenotype developed from seeds exposed to 4 h of heat wave at 40 °C in both Rayhane and Tborsek, compared to NT and the other treated samples.

### 3.2. Impact of Hydropriming on Germination Performance Under Heat Stress

Based on the preliminary characterization of germination performance under heat stress, the condition of 4 h of heat wave at 40 °C was selected to assess the impact of hydropriming on the seeds’ ability to withstand this adverse condition. As for the *L. sativus* accession Maleme-107, the heat stress treatment affected both the unprimed and primed seeds, triggering a similar decrease in germinability (Figure 5a). When MGT was evaluated, no significant difference was observed for the unprimed seeds in the presence/absence of the 40 °C heat wave. A similar response was evidenced for the hydroprimed seeds (Figure 5b). In Maleme-107, hydropriming resulted in an enhanced root length. However, growth was severely affected in response to heat stress in seedlings developed from both unprimed and hydroprimed seeds (Figure 5c). As for the *L. sativus* accession Sofades, the heat stress treatment affected both the unprimed and primed seeds, triggering a similar decrease in germinability (Figure 5d). When MGT was evaluated, a significant difference was observed for the unprimed seeds in the presence/absence of the 40 °C heat wave. However, the hydroprimed seeds exposed to the 40 °C heat wave did not show a significant difference in the estimated MGT, compared to the untreated hydroprimed seeds (Figure 5e). These findings suggest that a beneficial effect is associated with hydropriming, at least in terms of the germination speed. As reported for Maleme-107, hydropriming applied to Sofades resulted in an enhanced root length. However, growth was severely affected in response to heat stress in seedlings developed from both unprimed and hydroprimed seeds (Figure 5f).

In the case of the *P. sativum* var. *arvense* accession Forrimax, the heat stress treatment affected both the unprimed and primed seeds, triggering a significant decrease in germinability (Figure 6a). When MGT was evaluated, a significant germination delay was observed for the unprimed seeds when comparing the absence and presence of heat stress, and the same response was evidenced for the hydroprimed seeds (Figure 6b). In the *P. sativum* var. *arvense* accession Forrimax, no significant differences were observed in the root and shoot length of seedlings developed from unprimed and hydroprimed seeds. Conversely, heat stress induced a significant decrease in the shoot and root length of seedlings derived from unprimed as well as hydroprimed seeds (Figure 6c). As for Guifredo, the heat stress treatment severely affected both the unprimed and primed seeds. However, a limited but significant mitigation effect of hydropriming against heat stress was detected when evaluating the germination percentage (Figure 6d). When MGT was evaluated, a significant delay was observed for the unprimed seeds in the presence of the 40 °C heat wave. Similarly, the hydroprimed seeds exposed to the 40 °C heat wave showed a significant increase in MGT, compared to the untreated hydroprimed seeds. Moreover, hydropriming slightly but significantly improved the response to heat stress in terms of reduced MGT (Figure 6e). As reported for Forrimax, the application of hydropriming to Guifredo did not impact the root and shoot length, whereas heat stress significantly decreased the shoot and root length of seedlings developed from both unprimed and hydroprimed seeds (Figure 6f).

The *T. foenum-graecum* accession Rayhane revealed a comparable response to the heat stress treatment for both unprimed and primed seeds, with a significant decrease in germinability (Figure 7a). When MGT was evaluated, hydropriming significantly accelerated germination, while no significant difference was observed when comparing the absence and presence of heat stress in seedlings developed from unprimed as well as hydroprimed seeds (Figure 7b). In Rayhane, no significant differences were observed in the root and shoot length of seedlings developed from unprimed and hydroprimed seeds, whereas heat stress impaired the shoot and root length of seedlings developed from unprimed as well as hydroprimed seeds (Figure 7c). As for the *T. foenum-graecum* accession Tborsek, the heat stress treatment affected both the unprimed and hydroprimed seeds, triggering a similar decrease in germinability (Figure 7d). When MGT was evaluated, hydropriming significantly accelerated germination, while no significant difference was observed for the unprimed seeds in the presence/absence of the 40 °C heat wave. Similarly, the hydroprimed seeds exposed to the 40 °C heat wave did not show a significant difference in the estimated MGT, compared to the untreated hydroprimed seeds (Figure 7e). As reported for Rayhane, hydropriming applied to Tborsek did not impact the root and shoot length, while heat stress induced significant decreases in seedlings developed from unprimed as well as hydroprimed seeds (Figure 7f). The reported data did not consistently highlight significant differences in the germination profiles and seedling growth, which may correlate hydropriming with some protective effect against heat stress.

### 3.3. DNA Damage Patterns

An alkaline comet assay was used to assess the levels of DNA damage (single and double strand breaks) in embryos excised from the orphan legume seeds challenged with heat stress. As for Maleme-107, the 40 °C heat wave significantly increased the DNA damage in unprimed seeds (153.0 ± 3.6 a.u.), compared to the untreated control (131.8 ± 0.9 a.u.) (Figure 8a, UP-HW and UP-US). Hydropriming was able to significantly reduce DNA damage (114.4 ± 7.3 a.u.), compared to the unprimed sample (131.8 ± 0.9 a.u.) (Figure 8a, HP-US and UP-US). However, primed seeds challenged with heat stress did not show a beneficial impact, compared to the untreated primed seeds (Figure 8a, HP-HW and HP-US). A similar response was observed for Sofades; however, the amount of DNA damage detected in the hydroprimed seeds exposed to heat stress was significantly decreased (91.4 ± 9.6 a.u.), compared to the estimated value for the stressed unprimed seeds (123.7 ± 3.2 a.u.) (Figure 8b, HP-HW and UP-HW). As for Forrimax, no protective effect against genotoxic damage was highlighted in relation to hydropriming (Figure 8c). In the case of Guifredo, the amount of DNA damage observed in the hydroprimed seeds exposed to heat stress was significantly decreased (152.2 ± 6.2 a.u.), compared to the estimated value for the stressed unprimed seeds (164.4 ± 1.2 a.u.) (Figure 8d, HP-HW and UP-HW). Finally, both Rayhane and Tborsek accessions showed a significant increase in DNA damage in both unprimed and hydroprimed seeds exposed to the 40 °C heat wave, compared to their untreated counterparts (Figure 8e,f). The reported data suggest that hydropriming may possibly induce a protective effect against heat stress in *L. sativus* Sofades and *P. sativum* var. *arvense* Guifredo.

### 3.4. The Seed Antioxidant and Repair Response Triggered by Heat Stress in L. sativus Accessions

Based on the reported data, the *L. sativus* accessions Maleme-107 and Sofades were selected for an in-depth study of the seed repair response under heat stress, performed using *q*RT-PCR to assess the expression profiles of antioxidant and DDR (DNA damage response) genes. The analysis was focused on embryo axes excised from dry seeds (DSs), subjected to hydropriming and dry-back (DB), unprimed seeds subjected to 8 h of imbibition at 25 °C (UP-NT), unprimed seeds subjected to 4 h of imbibition at 25 °C and 4 h of heat wave at 40 °C (UP-HW), hydroprimed seeds subjected to 8 h of imbibition at 25 °C (HP-NT), and hydroprimed seeds subjected to 4 h of imbibition at 25 °C and 4 h of heat wave at 40 °C (HP-HW) (Figure 9). The relative expression data referring to the *q*RT-PCR results are provided in Appendix A (*L. sativus* accession Maleme-107) and Appendix A (*L. sativus* accession Sofades). The antioxidant response was investigated by selecting genes encoding the cytosolic isoform of superoxide dismutase and ascorbate peroxidase, both acting as key players in the seed pre-germinative response [33]. The tested antioxidant genes revealed expression patterns predominantly similar in the two *L. sativus* accessions. As for the *LsSODa* and *LsSODb* genes, when encoding the superoxide dismutase orthologs a and b identified in *L. sativus*, similar expression patterns were observed in Maleme-107 and Sofades. *LsSODa* and *LsSODb* transcript accumulation was detected in the dry seed as well as in unprimed seeds, in the presence and absence of the 40 °C heat wave. In contrast, lower transcript levels were observed after post-priming dry back and in hydroprimed seeds during re-imbibition, independent of heat stress. As for the *LsAPXa* and *LsAPXb* genes, when encoding the ascorbate peroxidase orthologs a and b identified in *L. sativus*, similar expression patterns were observed in Maleme-107 and Sofades. Low *LsAPXa* and *LsAPXb* transcript levels were detected in the dry seed, whereas peaks of expression were detected in response to post-priming dry back. The *LsAPXb* gene was, however, upregulated in response to heat stress in the unprimed seeds in both *L. sativus* accessions. In contrast, no or slight transcript accumulation was observed in hydroprimed seeds, independent of heat stress. Metallothioneins play protective roles against oxidative damage during seed imbibition [33]. The upregulation of *LsMT1* and *LsMT2* genes was predominant in dry and dried-back seeds of Maleme-107 (Figure 9a); meanwhile, in Sofades, the *LsMT1* and *LsMT2* genes were specifically upregulated in dried-back and dry seeds, respectively (Figure 9b). The *LsMT2* gene turned out to be upregulated in unprimed seeds in the presence/absence of heat stress in Sofades (Figure 9b). The *LsSPDS* and *LsSPMS* genes, involved in polyamines biosynthesis [34], were upregulated in both dried-back seeds and unprimed seeds in Maleme-107, in both stressed and unstressed seeds (Figure 9a). The *LsSOG1* gene, coding for the master regulator of plant DDR [35], displayed an accumulation peak in response to dry-back treatment in Maleme-107 seeds (Figure 9a). The expression of DNA repair genes was evaluated, including *LsOGG1*, *LsFPG* and *LsLig*; these encode DNA glycosylases and a DNA ligase involved in BER in multiple experimental systems including seeds [35,36]. The expression of *LsTop1*, encoding DNA topoisomerase I [37], *LsPCNA*, encoding proliferating cell nuclear antigen, which is involved in replication and DNA repair [38], and *LsTFIIS*, encoding an elongation cofactor of RNA polymerase II involved in plant heat stress response [39], was also evaluated. These six genes with roles in DNA repair displayed overall consistent expression patterns within the two *L. sativus* accessions, with higher transcript accumulation in unprimed seeds (Figure 9a,b); meanwhile, increases were detected in Sofades hydroprimed seeds responding to heat stress (Figure 9b, HP-HW). Exposure to heat stress was associated with overexpression in the case of *LsTop1* in the unprimed seeds of both accessions (Figure 9a,b, UP-HW) and in the case of *LsFPG*, *LsLig* and *LsPCNA* in Maleme-107 (Figure 9a, UP-HW). Regarding 5.8S rRNA, mature and precursor forms were reported to be potential hallmarks of seeds’ response to desiccation stress in *M. truncatula* and *M. sativa* [28]. The reported data show that the 5.8S rRNA mature form was accumulated in response to the dry-back treatment in both accessions, whereas contrasting profiles were observed in untreated seeds in the presence/absence of stress in the two accessions. In the case of the precursor 5.8S rRNA, accumulation was observed in the unprimed non-stressed seeds in Maleme-107 (Figure 9a, UP-NT) and in response to heat stress in hydroprimed seeds (HP-HW) in both accessions compared with the non-stressed hydroprimed seeds (HP-NT) (Figure 9a,b). The precursor 5.8S rRNA was also accumulated in the hydroprimed seeds in Sofades in the presence/absence of heat stress (Figure 9b).

### 3.5. Principal Component Analyses

The datasets for PCA analysis in the two tested *L. sativus* accessions were arranged to highlight the variations induced by hydropriming in combination with the heat wave treatment, resulting in four treatment groups for the two accessions: (i) unprimed non-stressed seeds (UP-NT), (ii) unprimed seeds subjected to heat wave (UP-HW), (iii) hydroprimed non-stressed seeds (HP-NT), and (iv) hydroprimed seeds subjected to heat wave (HP-HW). In both accessions, the replicated segregated into separate clusters, although with diverging configurations (Figure 10). In the Maleme-107 accession, the main driver of variability within the dataset along Principal Component (PC) 1 appeared to be the effect of hydropriming. Specifically, the separation of HP-NT samples was driven by variables referring to the germination percentage and seedling development (shoot and root length), and by the expression levels of *LsSOG1*, *LsAPX3a* and *LsMT1*. Conversely, the slower germination (higher mean germination time) and the higher expression levels of the remaining genes involved in the antioxidant and DDR response determined the clustering of unprimed samples (UP-NT and UP-HW). The variability induced by the heat wave configurated along PC2, with the accumulation of DNA strand breaks as the main driver (Figure 10a). In the Sofades accession, unprimed samples not subjected to heat wave (UP-NT) clustered close to the mean values at the origin of the PC1 and PC2 axis, with the treatments (UP-HW, HP-NT, and HP-HW) configurating diverging patterns. The clustering in response to hydropriming and no heat stress (HP-NT) appears to be driven by the germination percentage and seedling development (shoot and root length), as in Maleme-107; meanwhile, the slower germination, the higher expression levels of the remaining genes involved in the antioxidant and DDR response, and the accumulation of DNA strand breaks determined the clustering in response to the heat wave (UP-HW and HP-HW). Specifically, the expression levels of *LsMT1*, *LsAPX3a* and *LsSPDS* determined the clustering in response to hydropriming combined with the heat wave (HP-HW). The expression of a subset of genes (*LsAPX3b*, *LsSODb*, *LsMT2*, and *LsSPMS*) appears to be driving the response to the heat wave in unprimed seeds (UP-HW) in both accessions (Figure 10a,b).

## 4. Discussion

The present work explores the hypothesis that different accessions of commercially neglected legumes (*L. sativus* accessions Maleme-107 and Sofades, *P. sativum* var. *arvense* accessions Forrimax and Guifredo, and *T. foenum-graecum* accessions Rayhane and Tborsek) can show different tolerance profiles against heat stress occurring at early stages of seed germination. This is connected to the potential of underutilized ‘orphan’ legumes as stress-tolerant and climate-ready crops, adapted to agronomical systems of the Mediterranean area exposed to extreme drought and heat events. Such a deleterious impact on germination rates and seedling establishment constrains legume sowing to colder seasons [40]. In this study, heat stress (4 h at 40 °C) is applied to unprimed and hydroprimed seeds that are metabolically active (after 4 h of imbibition at 25 °C) to simulate a heat wave event and evaluate whether hydropriming is able to improve the seed response to heat stress. Heat wave is chosen since it resembles the temperature fluctuations observed in the daytime–night-time cycle along the coastlines of the Mediterranean area [41,42]. Other patterns of heat stress administration could be evaluated in future investigations, including continuous stress or combined stressors (e.g., drought stress). Several studies have explored different stress administration patterns adapted to different developmental stages. Heat waves (3 days, 6 °C above the ambient temperature recorded in the experimental field) applied to soybean (*Glycine max*) plants during pod development caused significant yield losses, despite its rapid recovery from oxidative stress [43]. Heat waves (continuous heat at 37 °C and 3 h of heat wave at 45 °C) applied to tomato (*Solanum lycopersicum*) seedlings significantly impacted seedling development, hormonal profiles, and the expression of genes encoding heat shock proteins [44].

In the present study, the germination and seedling development data highlight differential responses among the three tested species and among the selected accessions. Under the highest tested temperature (40 °C), the *L. sativus* accession Sofades displayed a higher germination percentage than Maleme-107, whereas the *P. sativum* var. *arvense* accession Guifredo was unable to germinate. When evaluating the intra- and inter-specific differences in stress response, it is crucial to establish a standardized experimental system; however, a high number of variables can also determine substantial differences in the stress response of seed lots from the same cultivar. A deeper investigation of the genetic and physiological factors contributing to seed vigor is necessary to provide mechanistic explanations from a comparative perspective, in order to complement the empirical approaches to testing stress tolerance [9]. Among these, seed age and the duration of the controlled imbibition treatment can influence the impact of heat stress and the response to priming [28,45].

The present work evaluated hydropriming as a resource-effective approach to improving the germination profiles of stressed and/or unstressed seeds of the target orphan legume accessions. Concerning unstressed seeds, hydropriming was able to significantly accelerate germination in all the tested accessions. Improved root development was detected in both *L. sativus* accessions. The observed positive effects of hydropriming are in agreement with the response observed in several crop and model species [9,46]. Hydropriming improved the germination speed (lower MGT) in both *L. sativus* accessions, in the *P. sativum* var. *arvense* accession Guifredo, and in both *T. foenum-graecum* accessions in both unstressed seeds and seeds subjected to heat stress, compared to their unprimed counterparts. The oreventive effects of hydropriming against heat stress were reported in multiple experimental systems, including *Pisum sativum* [8] and *Medicago sativa* [47]. These studies underline the potential ability of hydropriming to improve germination performance under extreme climate events.

Single-cell gel electrophoresis (comet assay) consistently showed an accumulation of DNA strand breaks in response to the heat wave in both primed and unprimed embryos. Nonetheless, lower DNA damage values were reported for hydroprimed seeds in the *L. sativus* accession Sofades, both in the presence and absence of heat stress. Slight but significant mitigation effects were highlighted also for the *L. sativus* accession Maleme-107 and the *P. sativum* var. *arvense* accession Guifredo. Based on the reported data, both *L. sativus* accessions were selected to investigate the expression of genes involved in the antioxidant response and DNA repair, and possibly highlight conserved or diverging responses related to DNA damage dynamics.

The relevance of antioxidant enzymes and compounds in preserving DNA integrity during germination has been reported in multiple experimental systems [48]. In the present study, the identified *L. sativus SOD* gene orthologs were less accumulated in hydroprimed seeds in both *L. sativus* accessions, with similar expression patterns. This downregulation suggests the lower requirements of the *LsSOD* gene functions. Conversely, in both accessions, the *L. sativus APX3* gene orthologs displayed different expression patterns, responding to dry-back or to heat stress in unprimed seeds. The two *L. sativus* accessions displayed a globally decreased expression of DDR genes (particularly *LsOGG1*, *LsFPG*, *LsLig*, *LsTop*, *LsTFIIS*, *LsPCNA*) in response to hydropriming. This decreased transcript accumulation may reflect a lower requirement of DDR components, consistent with the lower DNA damage levels detected in hydroprimed seeds. Nonetheless, in the Sofades accession, heat stress also induced the accumulation of DDR gene transcripts in hydroprimed seeds. A similar pattern was observed for the precursor and mature forms of 5.8S rRNA, which has been linked with a nucleolus-mediated stress response [28]. The expression of genes encoding antioxidant enzymes and DDR components has been reported in response to heat stress in seeds [49,50]. Although the use of gene expression analysis provided evidence that a subset of antioxidant and DDR genes is responsive to hydropriming and/or heat stress in *L. sativus* accessions, further study will be needed to elucidate their regulatory mechanisms. Gene expression analysis across priming and dry-back treatments must account for the notable differences in metabolic activity among dry seeds, seeds undergoing dry-back, and seeds undergoing imbibition, which can affect transcription and translation rates based on the progression of pre-germinative metabolism toward germination. The molecular patterns hereby provided will be integrated with the study of epigenetic regulators since it has been reported that histone modifications and chromatin changes occurring during exposure to short-term high temperature heat waves play a regulatory role [51]. When investigating seed thermotolerance, the contribution of other classes of proteins should be considered, including heat shock proteins [52]. Moreover, the assessment of the accumulation of reactive oxygen species levels, the activity of antioxidant enzymes, and non-enzymatic antioxidants should be considered to define the contribution of ROS homeostasis in relation to hydropriming, DNA integrity, and thermotolerance [53]. Finally, other priming approaches could be tested in future studies, including thermopriming [41,54,55]. Accounting for differences among species and cultivars is crucial to understanding the molecular and physiological bases of stress tolerance to boost the transition toward applicative directions for breeding programs and seed-invigorating strategies [9,44]. However, the complexity of seed physiology, the genetic background, and the influence of environment result in the enhanced variability of the seed response when novel techniques are applied, and require empirical optimization. Furthermore, the response of seeds to priming should be considered in view of crop susceptibility to the multiple environmental biotic and abiotic stresses that might occur concomitantly or subsequent to exposure to high temperatures [56]. It has been reported that heat stress accelerates deleterious physiological changes in wheat seedlings exposed to drought stress [53], whereas the combined heat and salinity stresses provided to rice trigger unique and complex combinations of molecular networks [57]. Current seed priming technologies would benefit from increased flexibility and efficacy, generated by integrating knowledge in seed biology/technology and the agronomy of orphan legumes, moving from bench to field. The mechanisms of heat sensing and downstream molecular responses can effectively trigger adaptation when a gradual temperature increase is applied [4,58]. Additional studies will be needed to test whether any gradual temperature increase will allow orphan legumes seeds to reinforce their capacity to adapt to heat stress.

The research findings hereby reported still need to be examined in the field to assess the effectiveness of hydropriming under dry and hot conditions. However, a study by Damalas et al. (2019) [59] showed the beneficial effect of hydropriming on faba bean plants under limited soil moisture, both at the early growth stage and after flowering. Moreover, Shrestha et al. (2024) [60] demonstrated the successful management of poor germination, yield issues, and inadequate plant protection measures in French bean (*Phaseolus vulgaris* L.) cultivation using hydropriming. Despite these promising results, there are still issues that need to be addressed in order to promote resilience in sustainable agrifood systems based on orphan legumes, including weather forecasting tools, the optimization of the mechanical harvesting of orphan legumes, funding tools, and farmers’ education on crop techniques, as highlighted by Goufa et al. (2024) [61].

## Figures and Tables

**Figure 1 genes-16-00235-f001:**
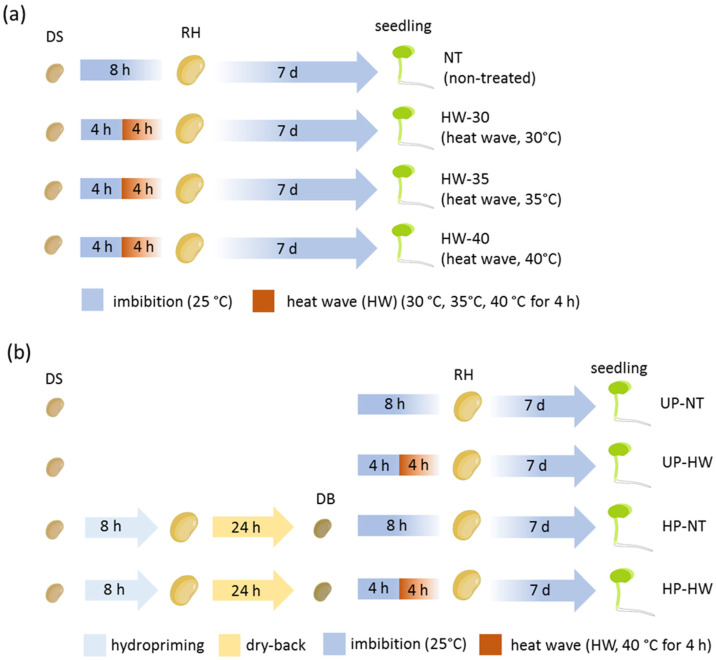
Overview of the experimental systems designed to assess the effects of heat waves on seeds of *L. sativus* accessions, *Pisum sativum* var. *arvense* accessions, and *Trigonella foenum-graecum* accessions. (**a**) Schematic representation of the experimental system designed to assess the impact of 4 h heat waves with incremental temperatures (25 °C, 30 °C, 35 °C, and 40 °C) administered after 4 h of imbibition at 24 °C. (**b**) Experimental system designed to assess the effects of 4 h of heat wave (40 °C) after 4 h of imbibition at 25 °C on unprimed and hydroprimed (8 h) seeds. DS, dry seed; DB, dry back; RH, rehydrated seed; UP, unprimed seeds; HP, hydroprimed seeds; NT, seeds subjected to 8 h of imbibition at 25 °C without heat wave; HW, seeds subjected to 4 h of imbibition at 25 °C followed by 4 h of heat wave at 40 °C.

**Figure 2 genes-16-00235-f002:**
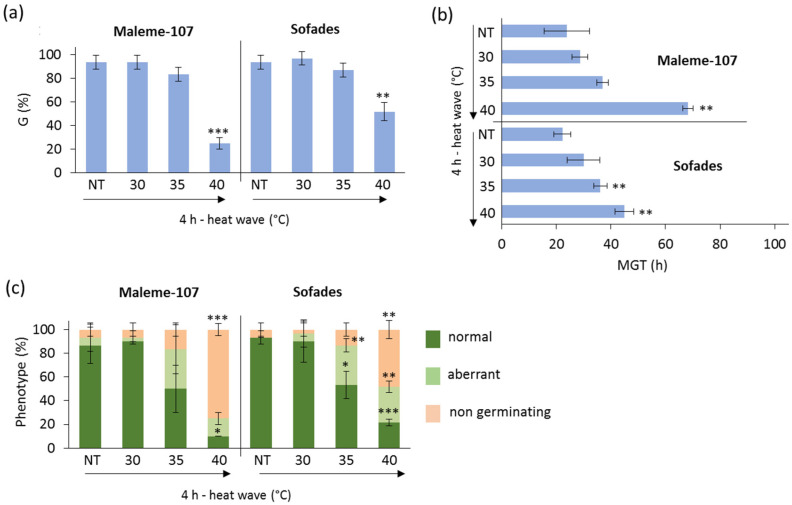
Germination performance of the *L. sativus* accessions Maleme-107 and Sofades under control and heat wave conditions. (**a**) Germinability index (G). (**b**) Mean germination time (MGT). (**c**) Frequency of seedlings with normal and aberrant phenotypes developed from control and treated seeds. NT, non-treated with heat wave; HW, heat wave. Asterisks indicate statistically significant differences in each heat wave condition compared with the respective non-treated control, as analyzed using a heteroscedastic two-tailed Student’s *t*-test. *, *p*-value < 0.05; **, *p*-value < 0.01; ***, *p*-value < 0.001.

**Figure 3 genes-16-00235-f003:**
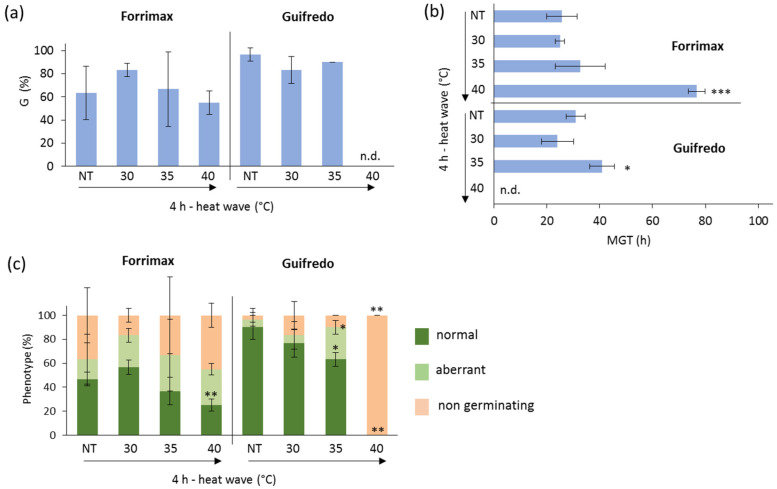
Germination performance of the *P. sativum* var. *arvense* accessions Forrimax and Guifredo under control and heat wave conditions. (**a**) Germinability index (G). (**b**) Mean germination time (MGT). (**c**) Frequency of seedlings with normal and aberrant phenotypes developed from control and treated seeds. NT, non-treated with heat wave; HW, heat wave. Asterisks indicate statistically significant differences in each heat wave condition compared with the respective non-treated control, as analyzed using a heteroscedastic two-tailed Student’s *t*-test. *, *p*-value < 0.05; **, *p*-value < 0.01; ***, *p*-value < 0.001.

**Figure 4 genes-16-00235-f004:**
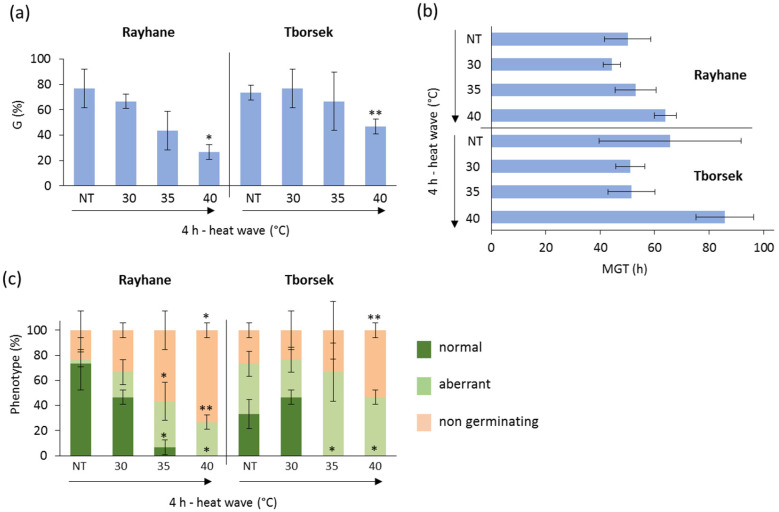
Germination performance of the *T. foenum-graecum* accessions Rayhane and Tborsek under control and heat wave conditions. (**a**) Germinability index (G). (**b**) Mean germination time (MGT). (**c**) Frequency of seedlings with normal and aberrant phenotypes developed from control and treated seeds. NT, non-treated with heat wave; HW, heat wave. Asterisks indicate statistically significant differences in each heat wave condition compared with the respective non-treated control, as analyzed using a heteroscedastic two-tailed Student’s *t*-test. *, *p*-value < 0.05; **, *p*-value < 0.01; ***, *p*-value < 0.001.

**Figure 5 genes-16-00235-f005:**
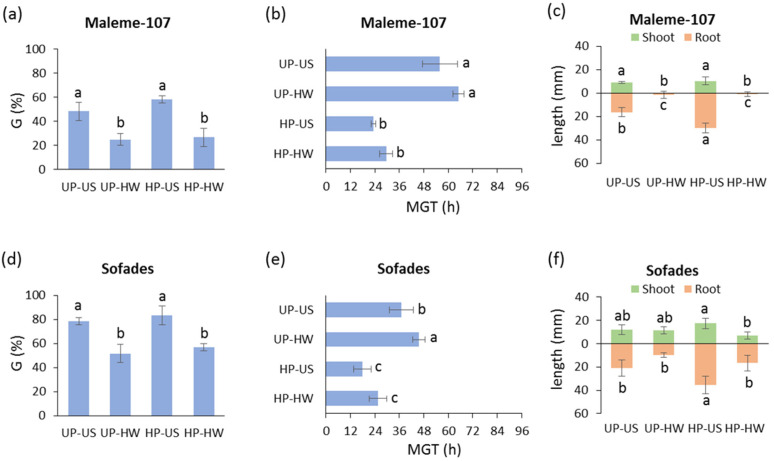
Germination performance of the *L. sativus* accessions Maleme-107 and Sofades (unprimed, hydroprimed) under control and heat wave conditions. (**a**) Germinability index in *L. sativus* Maleme-107 accession. (**b**) Mean germination time in *L. sativus* Maleme-107 accession. (**c**) Shoot and root length in 7-day-old seedlings in *L. sativus* Maleme-107 accession. (**d**) Germinability index in *L. sativus* Sofades accession. (**e**) Mean germination time in *L. sativus* Sofades accession. (**f**) Shoot and root length in 7-day-old seedlings in *L. sativus* Sofades accession. UP, unprimed seeds; HP, hydroprimed seeds; US, unstressed seeds (subjected to 8 h of imbibition at 25 °C); HW, heat wave (seeds subjected to 4 h of imbibition at 25 °C followed by 4 h of heat wave at 40 °C). G, germinability index; MGT, mean germination time. Values are indicated as average ± standard deviation. For each parameter of each accession, average values without common letters are significantly different (*p*-value < 0.05), as analyzed by one-way ANOVA and Duncan’s test.

**Figure 6 genes-16-00235-f006:**
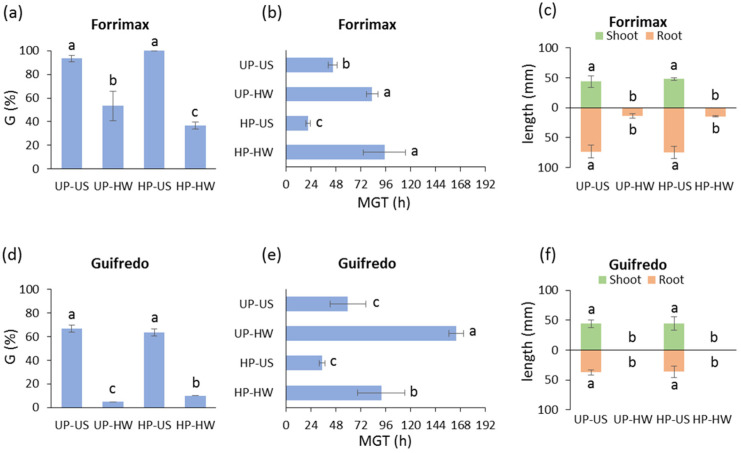
Germination performance of the *P. sativum* var. *arvense* accessions Forrimax and Guifredo (unprimed, hydroprimed) under control and heat wave conditions. (**a**) Germinability index in *P. sativum* var. *arvense* Forrimax accession. (**b**) Mean germination time in *P. sativum* var. *arvense* Forrimax accession. (**c**) Shoot and root length in 7-day-old seedlings in *P. sativum* var. *arvense* Forrimax accession. (**d**) Germinability index in *P. sativum* var. *arvense* Guifredo accession. (**e**) Mean germination time in *P. sativum* var. *arvense* Guifredo accession. (**f**) Shoot and root length in 7-day-old seedlings in *P. sativum* var. *arvense* Guifredo accession. UP, unprimed seeds; HP, hydroprimed seeds; US, unstressed seeds (subjected to 8 h of imbibition at 25 °C); HW, heat wave (seeds subjected to 4 h of imbibition at 25 °C followed by 4 h of heat wave at 40 °C). G, germinability index; MGT, mean germination time. Values are indicated as average ± standard deviation. For each parameter of each accession, average values without common letters are significantly different (*p*-value < 0.05), as analyzed by one-way ANOVA and Duncan’s test.

**Figure 7 genes-16-00235-f007:**
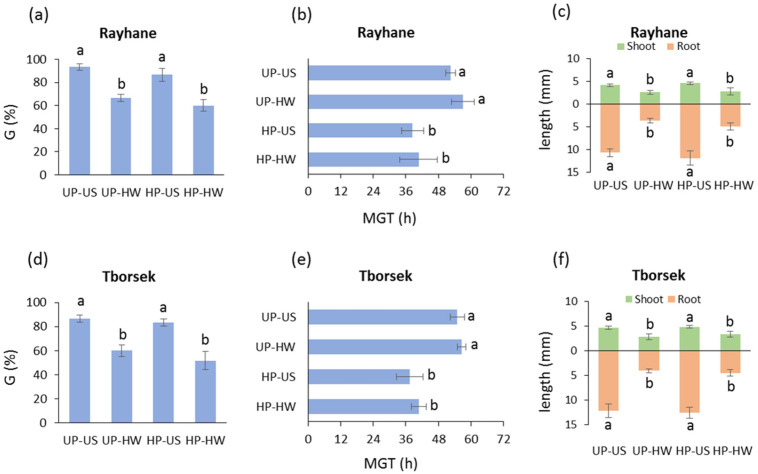
Germination performance of the *T. foenum-graecum* accessions Rayhane and Tborsek (unprimed, hydroprimed) under control and heat wave conditions. (**a**) Germinability index in *T. foenum-graecum* Rayhane accession. (**b**) Mean germination time in *T. foenum-graecum* Rayhane accession. (**c**) Shoot and root length in 7-day-old seedlings in *T. foenum-graecum* Rayhane accession. (**d**) Germinability index in *T. foenum-graecum* Tborsek accession. (**e**) Mean germination time in *T. foenum-graecum* Tborsek accession. (**f**) Shoot and root length in 7-day-old seedlings in *T. foenum-graecum* Tborsek accession. UP, unprimed seeds; HP, hydroprimed seeds; US, unstressed seeds (subjected to 8 h of imbibition at 25 °C); HW, heat wave (seeds subjected to 4 h of imbibition at 25 °C followed by 4 h of heat wave at 40 °C); G, germinability index; MGT, mean germination time. Values are indicated as average ± standard deviation. For each parameter of each accession, average values without common letters are significantly different (*p*-value < 0.05), as analyzed by one-way ANOVA and Duncan’s test.

**Figure 8 genes-16-00235-f008:**
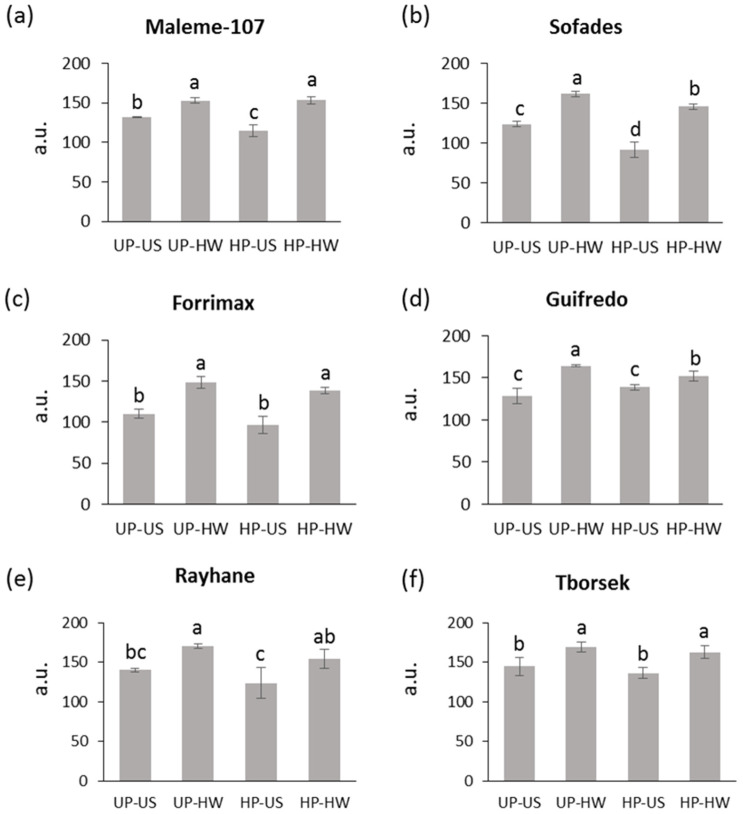
DNA damage (single and double strand breaks) levels in embryos excised from unprimed and hydroprimed orphan legume seeds exposed to 4 h of heat wave at 40 °C. (**a**) *L. sativus* accession Maleme-107. (**b**) *L. sativus* accession Sofades. (**c**) *P. sativum* var. *arvense* accession Forrimax. (**d**) *P. sativum* var. arvense accession Guifredo. (**e**) *T. foenum-graecum* accession Rayhane. (**f**) *T. foenum-graecum* accession Tborsek. UP, unprimed seeds; HP, hydroprimed seeds; US, unstressed seeds (subjected to 8 h of imbibition at 25 °C); HW, heat wave (seeds subjected to 4 h of imbibition at 25 °C followed by 4 h of heat wave at 40 °C); a.u., arbitrary units. Values are indicated as average ± standard deviation. For each accession, average values without common letters are significantly different (*p*-value < 0.05), as analyzed by one-way ANOVA and Duncan’s test.

**Figure 9 genes-16-00235-f009:**
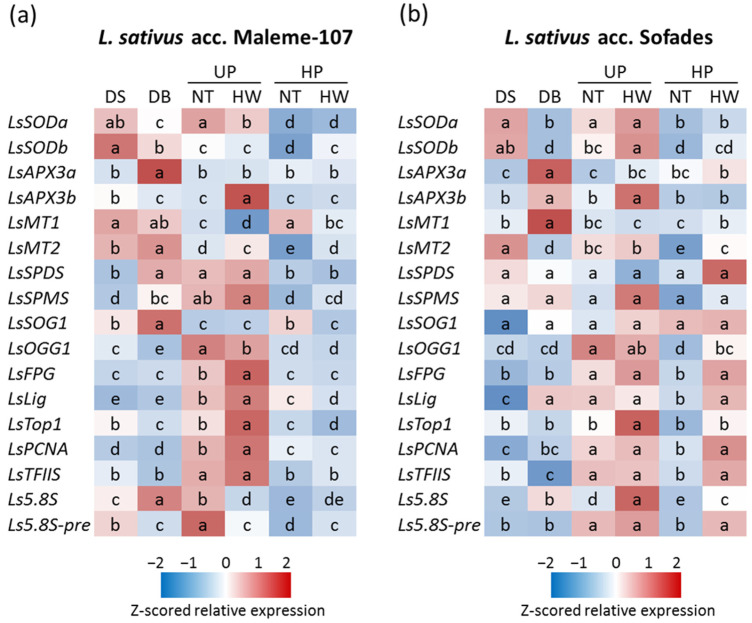
Heat maps representing the relative expression profiles of genes involved in the seed repair response, evaluated by *q*RT-PCR in *L. sativus* accession. (**a**) Z-scored relative gene expression in *L. sativus* Maleme-107. (**b**) Z-scored relative gene expression in *L. sativus* Sofades. For each gene of each accession, expression levels without common letters are significantly different (*p*-value < 0.05), as analyzed by one-way ANOVA and Duncan’s test. DS, dry seed; DB, seeds subjected to hydropriming and dry-back; UP, unprimed seeds; HP, hydroprimed seeds; NT, seeds subjected to 8 h of imbibition at 25 °C without heat wave; HW, seeds subjected to 4 h of imbibition at 25 °C followed by 4 h of heat wave at 40 °C. *LsSOD*, superoxide dismutase. *LsAPX*, ascorbate peroxidase. *LsMT*, metallothionein. *LsSPMS*, spermine/spermidine synthase. *LsSPDS*, spermidine synthase. *LsSOG1*, suppressor of the gamma response *1. LsOGG1*, 8-oxoguaninglycosylase/lyase. *LsFPG*, formamidopyrimidine-dna glycosylase. *LsLig*, DNA ligase. *LsTOP*, DNA topoisomerase. *LsPCNA*, proliferating cell nuclear antigen. *LsTFIIS*, transcription elongation factor *IIS. Ls5.8S*, *5.8S rRNA. Ls5.8S-IS*, *5.8S rRNA-*interspace. *Ls*, *Lathyrus sativus*.

**Figure 10 genes-16-00235-f010:**
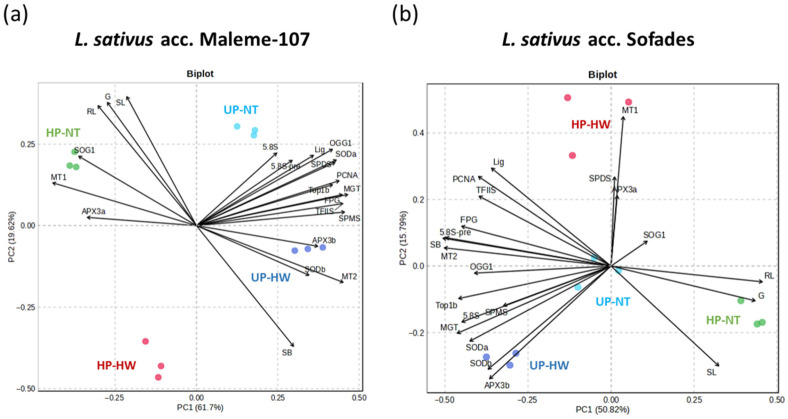
Principal component analysis of the effects of heat stress on unprimed and hydroprimed *L. sativus* seeds. (**a**) Biplot referring to *L. sativus* accession Maleme-107. (**b**) Biplot referring to *L. sativus* accession Sofades. *SOD*, superoxide dismutase. *APX*, ascorbate peroxidase. *MT*, metallothionein. *OGG1*, 8-oxoguaninglycosylase/lyase. *FPG*, formamidopyrimidine-dna glycosylase. *Lig*, *DNA* ligase. *TOP*, *DNA* topoisomerase. *TFIIS*, transcription elongation factor *IIS. SPMS*, *SPERMINE/SPERMIDINE SYNTHASE. SPDS*, spermidine synthase. *5.8S*, *5.8S rRNA. 5.8S-IS*, *5.8S rRNA-*interspace. *PCNA*, proliferating cell nuclear antigen. *SOG1*, suppressor of the γ response 1. G, germinability. MGT, mean germination time. SL, shoot length. RL, root length. SB, strand break.

## Data Availability

All the research data generated in the present work are presented in the manuscript.

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
