# Peer review of "Exploring the Genotoxic Stress Response in Primed Orphan Legume Seeds Challenged with Heat Stress"

_genes, 2025, doi:10.3390/genes16020235_

Round 1
Reviewer 1 Report
Comments and Suggestions for Authors
This study investigates the effects of hydropriming as a mitigation strategy against heat stress in orphan legume seeds, focusing on DNA damage and repair mechanisms. Using germination performance metrics, comet assays, and gene expression analysis, the authors evaluate the impact of hydropriming on thermotolerance. The work contributes valuable insights into the genetic and physiological mechanisms underpinning seed vigour under climate stress conditions.
Comment 1: Is there a need to discuss other abiotic stresses (e.g., drought, salinity) that may interact with heat stress in real-world conditions?
Comment 2: How were the selected temperatures (30°C, 35°C, and 40°C) determined, and could a wider range be tested for better threshold identification?
Comment 3: Could additional controls, such as unprimed seeds subjected to a gradual temperature increase, strengthen the conclusions?
Comment 4: Were there any potential confounding factors (e.g., seed age, hydration status) that might affect hydropriming efficiency?
Comment 5: Could gene expression analysis be extended to include epigenetic regulators that might play a role in heat stress resilience?
Comment 6: How do these findings translate to field conditions, and what additional environmental factors need consideration?
Comment 7: What practical recommendations could be derived for farmers or agronomists to apply hydropriming more effectively?
Author Response
Please, see attached file

Reviewer 2 Report
Comments and Suggestions for Authors
The paper contains interesting results on the possibility of using hydropriming to reduce heat stress on seed germination of faba bean plants (Lathyrus sativus L., Pisum sativum var. arvense, Trigonella foenum-graecum L.). The paper relates very well to the problem of accelerating climate change. I ask the Authors to be more explicit about the applicability of their research results in breeding new varieties, agricultural production. What obstacles do they see? Which problems still need to be solved quickly?
Introduction
It is too lengthy and needs to be shortened.
Materials and Methods
Where was the research carried out? In which year?
Please complete the manuscript with a brief description of the varieties studied.
Please provide full details of the producer of the statistical software.
Discussion
„Authors should discuss the results and how they can be interpreted from the perspective of previous studies and of the working hypotheses. The findings and their implications should be discussed in the broadest context possible. Future research directions may also be highlighted”. ???
I think this is an excerpt from another review, and I agree with it entirely.
References
Please use publications from the last 10 years and delete the others. Exceptions are publications necessary to characterise the research methodology.
